# Emerging Prognostic Markers in Patients Undergoing Liver Resection for Hepatocellular Carcinoma: A Narrative Review

**DOI:** 10.3390/cancers16122183

**Published:** 2024-06-10

**Authors:** Elena Panettieri, Andrea Campisi, Agostino M. De Rose, Caterina Mele, Felice Giuliante, Jean-Nicolas Vauthey, Francesco Ardito

**Affiliations:** 1Department of Surgical Oncology, The University of Texas MD Anderson Cancer Center, Houston, TX 77030, USA; jvauthey@mdanderson.org; 2Hepatobiliary Surgery, Fondazione Policlinico Universitario A. Gemelli IRCCS, Università Cattolica del Sacro Cuore, 00168 Rome, Italy; andrea.campisi03@icatt.it (A.C.); agostinomaria.derose@policlinicogemelli.it (A.M.D.R.); caterina.mele@policlinicogemelli.it (C.M.); felice.giuliante@unicatt.it (F.G.); francesco.ardito@unicatt.it (F.A.)

**Keywords:** hepatocellular carcinoma, liver resection, liquid biopsy, biomarkers, prognostic markers

## Abstract

**Simple Summary:**

Hepatocellular carcinoma (HCC) is a primary tumor of the liver with a particularly high incidence in Asia. HCC has a tendency to recur despite curative-intent liver resection, and new reliable prognostic markers would have high clinical relevance. Liquid biopsy markers, gene signatures, and inflammation markers represent valuable tools to predict recurrence and overall survival in patients undergoing hepatectomy for HCC. Nevertheless, while some of the proposed new markers have been widely investigated and demonstrated to be reproducible, others seem less reproducible in clinical practice. In particular, liquid biopsy is a powerful tool for predicting long-term outcomes after resection of HCC; with costs related to its technical implementation representing a major limitation. More reports from Western countries are necessary to corroborate the evidence.

**Abstract:**

In patients with hepatocellular carcinoma (HCC), liver resection is potentially curative. Nevertheless, post-operative recurrence is common, occurring in up to 70% of patients. Factors traditionally recognized to predict recurrence and survival after liver resection for HCC include pathologic factors (i.e., microvascular and capsular invasion) and an increase in alpha-fetoprotein level. During the past decade, many new markers have been reported to correlate with prognosis after resection of HCC: liquid biopsy markers, gene signatures, inflammation markers, and other biomarkers, including PIVKA-II, immune checkpoint molecules, and proteins in urinary exosomes. However, not all of these new markers are readily available in clinical practice, and their reproducibility is unclear. Liquid biopsy is a powerful and established tool for predicting long-term outcomes after resection of HCC; the main limitation of liquid biopsy is represented by the cost related to its technical implementation. Numerous patterns of genetic expression capable of predicting survival after curative-intent hepatectomy for HCC have been identified, but published findings regarding these markers are heterogenous. Inflammation markers in the form of prognostic nutritional index and different blood cell ratios seem more easily reproducible and more affordable on a large scale than other emerging markers. To select the most effective treatment for patients with HCC, it is crucial that the scientific community validate new predictive markers for recurrence and survival after resection that are reliable and widely reproducible. More reports from Western countries are necessary to corroborate the evidence.

## 1. Introduction

Hepatocellular carcinoma (HCC) is globally ranked as the third leading cause of cancer-related death, and the incidence of HCC is particularly high in eastern and southeastern Asia [1]. The main therapeutic options for HCC are liver resection (LR), radiofrequency or microwave ablation, and liver transplant, with the most appropriate treatment depending on tumor burden, liver function, and performance status [2,3].

LR in well-selected candidates leads to the best outcomes and can even be an option for patients with more advanced HCC [3]. Nevertheless, the rate of recurrence after LR is up to 70% at 5 years, and recurrence is associated with poor outcomes; thus, close follow-up after LR is recommended [4,5]. At present, follow-up after LR is based on liver sonography combined with measurement of alpha-fetoprotein (AFP) level [6]. However, this approach is unsatisfactory because the sensitivity of sonography in the detection of recurrent HCC is only 63%—and even lower in patients with cirrhosis—and the specificity of AFP in the detection of recurrent HCC is poor [7].

To guide medical and surgical treatments, there is an urgent need to identify novel predictors of recurrence and survival after LR for HCC. In the past decade, many new biomarkers have been reported to correlate with the prognosis of patients with HCC undergoing LR. Specifically, three categories of markers have been identified: (i) markers measured in liquid biopsy specimens, (ii) gene signatures, and (iii) markers associated with inflammation. A few markers not in any of these categories have also been identified, including prothrombin induced by vitamin K absence-II (PIVKA-II), programmed cell death protein-1 (PD-1), microRNAs, and proteins in urinary exosomes (Figure 1) [8,9,10,11].

Unfortunately, not all of the aforementioned markers are readily available in clinical practice, and their reproducibility is unclear. The aim of this review is to summarize the use of emerging biomarkers in predicting recurrence and survival after LR for HCC and provide clinicians with a snapshot of the current scientific evidence.

## 2. Liquid Biopsy Markers

Liquid biopsy, the collection of tumor material from body fluids such as circulating tumor cells (CTCs), circulating tumor DNA (ctDNA), and circulating microparticles (CMs) secreted by viable cancer cells, represents a minimally invasive and reproducible tool to acquire molecular data to predict the risk of recurrence and death after LR for HCC [12,13] (Figure 2).

### 2.1. CTC Level

Epithelial-mesenchymal transition (EMT) is a multistep, reversible process in which epithelial cells de-differentiate and acquire a mesenchymal phenotype. EMT allows tumor cells to enter the bloodstream and become CTCs [14]. CTSs are defined as nucleated cells positive for epithelial cell adhesion molecule (EpCAM) and cytokeratin and negative for CD45. CTCs are scarce and usually get rapidly damaged by shear stress, killed by the immune system, or apoptosis [15,16]. Only a small proportion of CTSs can survive and nestle in peripheral tissue, from which they originate local and distant metastases. Such early hematogenous spread of HCC is associated with increased risk for early recurrence and poor prognosis and should be detected in the perioperative period with high-sensitivity methods [17].

Detection of CTCs presents technical difficulties and is accomplished using one of three main techniques: (i) use of antibodies against cell surface markers (CellSearch and CTC-Chip assays), (ii) exploitation of cell biophysical properties (e.g., membrane filtration, dielectric mobility, inertial focusing, and dielectric mobility), and (iii) enrichment-free methods (flow cytometry) [16]. At present, CellSearch is the only U.S. Food and Drug Administration-approved CTC detection tool.

In a 2013 report, Sun et al. [18] demonstrated that a preoperative CTC count of ≥2 CTCs/7.5 mL predicted tumor recurrence after surgery in 123 HCC patients, especially in patients with AFP levels ≤400 ng/mL or low tumor recurrence risk. Later, the same group divided HCC patients who underwent LR into a retrospective training cohort (144 patients) and a prospective validation cohort (53 patients) and showed that a CTC count ≥3 CTCs/7.5 mL after LR was associated with a higher risk of extrahepatic metastases and reduced survival [19]. Similar findings were observed in other studies that utilized immunoaffinity technologies for CTC detection [20,21,22].

Fan et al. [23] performed a prospective study in which multicolor flow cytometry was used to count CTCs in serum samples of 82 HCC patients the day before LR. The authors observed that HCC patients with a CTC percentage (the proportion of examined cells that are CTCs) >0.01% were at higher risk of intrahepatic and extrahepatic recurrence and had a lower overall survival (OS) rate compared with patients with a CTC percentage ≤0.01%. A multivariable analysis showed that CTC percentage >0.01% was an independent predictor of poor disease-free survival (DFS). Hamaoka et al. [24] used multicolor flow cytometry to detect CTCs in the bloodstream of 85 HCC patients before LR and found a median of 3 CTCs/8 mL per patient. Interestingly, this study proved that CTC count of ≥5 CTCs/8 mL was associated with a higher risk of microscopic portal vein invasion, lower DFS, and lower OS.

Similarly, Ha et al. [25] performed a prospective study in which 105 HCC patients had blood sampling before and after LR was performed utilizing a tapered slit platform. An increase in CTCs after LR was significantly associated with a higher risk of recurrence, and subgroup analysis showed that an increase in CTCs was also related to lower OS among patients with cirrhosis.

Notably, the ongoing FINDIN-GIBIOREC study (clinicaltrials.gov ID NCT04800497) is a prospective observational cohort study designed to identify the early presence of CTCs in HCC patients who underwent LR. This multicenter protocol utilizes FACSymphony cell analyzers (BD Biosciences, Freemont, CA, USA) to detect the CTC markers EpCAM, neural cadherin, and CD90 in the perioperative period and integrates information about these markers with findings on computed tomography, magnetic resonance imaging, and AFP level to demonstrate a correlation with recurrence and cancer-related prognosis.

Table 1 summarizes findings from studies assessing the relationship between CTCs and prognosis after LR for HCC.

### 2.2. CTC Subtype

Epithelial cells usually show apical–basal polarity and are held together laterally by tight junctions and adherens junctions, which maintain the structure of epithelia. EMT allows epithelial cells to lose their polygonal shape and acquire a spindle-shaped mesenchymal morphology [26]. This transition is associated with the repression of surface epithelial cadherin expression in favor of the expression of mesenchymal markers such as neural cadherin, vimentin, and fibronectin [27]. EMT and the reverse process, mesenchymal-epithelial transition, are involved in HCC progression and recurrence, giving tumor cells metastatic potential [28]. Consequently, many authors have focused on markers of EMT and mesenchymal-epithelial transition on CTCs to develop high-sensitivity systems to predict recurrence and OS in patients with HCC [29,30,31,32,33,34,35].

In particular, Qi et al. [32] aimed to detect and classify CTCs in the bloodstream before LR. A CTC count ≥ 16 and a mesenchymal-CTC percentage ≥ 2% were found to correlate with ER, multiple intrahepatic recurrences, and lung metastasis. The authors discovered an evident upregulation of the *BCAT1* gene that seemed to regulate EMT. The same group performed a retrospective study using the same technique to classify CTCs in 136 HCC patients undergoing LR with negative margins [32]. They discovered that the preoperative presence of a high CTC count and the presence of mesenchymal CTC phenotypes were associated with extrahepatic and disseminated intrahepatic recurrence with low DFS. Innovatively, these authors suggested that anatomical resection may be advantageous in patients with low CTC count, and mesenchymal- and epithelial/mesenchymal-negative phenotypes in order to achieve a potentially curative operation. Prognostic benefits and surgical risks should be carefully balanced to plan the optimal resection approach.

Interestingly, CanPatrol was also used by Chen et al. [35] to count and classify CTCs with EMT markers in HCC patients before and after LR. However, in this study, no correlation was described between mesenchymal CTC count and clinical stage or ER.

Court et al. found that vimentin-positive CTCs can not only help diagnose and stage HCC but also accurately discriminate patients eligible for liver transplant (median, 0 vimentin-positive CTCs) from patients with locally advanced/metastatic disease ineligible for liver transplant (median, 6 vimentin-positive CTCs). Vimentin-positive CTCs can also predict earlier recurrence after curative-intent surgical or locoregional therapy in potentially curable early-stage HCC [36].

Table 2 summarizes findings from studies assessing the relationship between CTC subtypes and prognosis after LR for HCC.

Orrapin et al. [37] performed a systematic review of studies that examined CTCs expressing EMT markers (EMT-CTCs), circulating cancer stem cells, or both in patients with HCC. These authors found that most of the studies confirmed a correlation between the presence of EMT-CTCs and circulating cancer stem cells in the bloodstream and other negative prognostic factors: larger tumor size, advanced stage, vascular invasion, distant metastases, and ER.

### 2.3. Circulating Tumor DNA

ctDNA is the fraction of cell-free DNA (cfDNA) that can be detected in the bloodstream. ctDNA is released from dead tumor cells and from macrophages that have phagocytized tumor cells [16,38]; ctDNA can also be actively secreted by tumors as a signaling molecule [39]. The average concentration of cfDNA in cancer patients is 180 ng/mL and ctDNA usually constitutes approximately 0.01% of total cfDNA [40]. ctDNA maintains the genetic mutations and epigenetic changes of the originating tumor cells, andit started being considered as a biomarker for tumor diagnosis and monitoring of disease and response to therapy during the last decade [41].

ctDNA’s short half-life and stability in the bloodstream in theory would allow ctDNA to be a highly sensitive predictor of recurrence and an accurate prognostic factor in HCC patients who underwent LR [42]. However, the utility of ctDNA in clinical practice is limited. It can be challenging to distinguish ctDNA from cfDNA originating from different body tissues. In addition, ctDNA is thought to be released more easily by less aggressive strains of tumor cells, which means that ctDNA analysis might not reveal information regarding more aggressive tumor cell populations [16,43].

Different kits are available for ctDNA extraction, but the current techniques lack standardization. The first approach to ctDNA detection is based on targeting one or a few tumor-specific mutations known from the primary tumor; alternative approaches are to perform a genomewide analysis for copy number aberrations or whole genome sequencing [39].

Corcoran et al. [44] illustrated the role of ctDNA in monitoring response to cancer treatment and showed how the ctDNA level decreased after LR in HCC patients, implying the importance of monitoring the trends of ctDNA levels in the blood. Tokuhisa et al. [45] and An et al. [46] confirmed that ctDNA level in serum predicted prognosis after LR for HCC in terms of OS, DFS, ER, and development of extrahepatic metastases. Cai et al. [47] demonstrated how ctDNA might be routinely used as a dynamic biomarker for HCC recurrence after curative-intent hepatectomy. They proved that serial ctDNA sampling performed better in HCC surveillance than did protein biomarkers such as AFP, AFP-L3 (an isoform of AFP), and des-gamma-carboxy prothrombin, identifying microscopic residual tumors and revealing recurrence before it was revealed by imaging.

Liao et al. [48] utilized the MiSeq system to highlight *TERT*, *CTNNB1*, and *TP53* gene mutations in ctDNA and correlated their presence with the clinical outcome of HCC patients who underwent LR. The authors suggested that tumor-associated ctDNA mutations correlate with vascular invasion and shorter DFS in HCC patients.

Shen et al. [49] divided 895 HCC patients into three cohorts and demonstrated that the *TP53* R249S mutation in ctDNA, detected with droplet digital polymerase chain reaction (PCR), represents a promising prognostic biomarker for HCC patients, regardless of whether they have undergone LR. Among patients who had undergone LR (275 patients), patients with the R249S mutation had worse OS and DFS than patients without this mutation.

Another interesting potential ctDNA-related biomarker is the level of methylation or demethylation of ctDNA gene sequences, which seems to provide an early warning of residual HCC and ER after hepatectomy. For instance, Liu et al. [50] found that *LINE1* hypomethylation and *RASSF1A* promoter hypermethylation predict worse OS and ER. Additionally, Chan et al. [51] demonstrated that residual HCC after LR is related to the extent of ctDNA demethylation. A clinical trial by Xu et al. [52] showed the importance of HCC methylation markers in surveillance and prognostication for patients with HCC and established a prognostic prediction model based on a 10-marker panel. Furthermore, these authors built a prognostic score (cd-score) that was validated as an independent prognostic risk factor and appeared to be superior to AFP for predicting prognosis.

Table 3 summarizes findings from studies assessing the relationship between ctDNA and ctDNA mutations and the prognosis after LR for HCC.

### 2.4. Circulating Microparticles

Another interesting biomarker that can be measured in liquid biopsy specimens is CMs, which are extracellular vesicles with a diameter of 100 nm to 1000 nm [53]. CMs are produced through the budding of the cellular membrane and released in the bloodstream, and they are consequently marked with the same surface antigens as the originating cells. CMs are involved in horizontal communication between cells, can allow for the detection of liver malignancies, and predict a high risk of HCC recurrence after hepatectomy [54].

Abbate et al. [55] utilized flow cytometry to identify CMs expressing hepatocyte paraffin 1 (HepPar1) in the bloodstream of HCC patients, and these authors found that CM concentration was higher than in healthy patients. The authors also demonstrated that CM concentration before LR was significantly higher in HCC patients who developed ER than in those with no recurrence, suggesting that CMs expressing HepPar1 are a predictor of HCC recurrence after LR.

Furthermore, several studies have shown that circulating exosomal noncoding RNA (microRNA), often released from necrotic tumor cells, might represent a novel prognostic biomarker in HCC patients. Liu et al. [56], Shi et al. [57], and Tian et al. [58] described several microRNAs (miR-638, miR-125b, miR-21, and miR-10b) whose exosomal levels were associated with negative prognostic factors, such as larger tumor size and number, vascular invasion, advanced TNM stage, and poor prognosis. Reduced concentrations of miR-638 in CMs (detected with real-time PCR) in HCC patients after LR were associated with poorer OS, suggesting exosome-delivered miR-638 as an emerging biomarker to predict HCC prognosis [56].

Notably, Luo et al. [59] used quantitative PCR to measure exosomal circular RNA AKT3 (circAKT3) in serum samples of patients with HCC who had undergone LR and found that levels were significantly higher in patients with ER and poor OS.

Table 4 summarizes findings from studies assessing the relationship between CMs and prognosis after LR for HCC.

## 3. Gene Signatures

Pathological factors, including tumor extension, microvascular invasion, and lymph node involvement, have been widely recognized as predictors of recurrence and survival after LR for HCC [60], but the molecular mechanisms underlying HCC recurrence remain unclear.

Notably, gene signatures, including autophagy-related gene signatures, are emerging from molecular profiling of HCC [61,62,63]. This seems particularly relevant since the identification of commonly altered pathways has historically been more challenging for HCC than for other solid malignancies.

A gene signature consists of a single gene or a group of genes in a cell having a unique pattern of expression associated with a specific phenotype or outcome [64]. RNA sequencing and microarray are among the most commonly used technologies for transcriptome profiling and the identification of characteristic molecular gene signatures [61].

Several genetic mutations and gene expression patterns have been reported to be excellent tools to predict HCC recurrence and OS after LR for HCC [61,65,66,67,68,69,70,71]. Notably, He et al. [67] found that gene rearrangement in HCC was frequently associated with low tumor differentiation, tumor necrosis, microvascular invasion, elevated AFP levels, and mutations in *TP53*, *NTRK3*, and *BRD9*. These authors found that cumulative DFS rates at 1, 2, and 3 years after LR in patients with HCC were 45.1%, 31.9%, and 31.9%, respectively, for patients with gene rearrangement and 72.5%, 57.9%, and 49.0%, respectively, for patients without gene rearrangement. Furthermore, gene rearrangement appeared to be an independent risk factor for lower DFS.

In one of the largest published studies of gene expression and prognosis in patients with HCC, Wang et al. [70] matched the mRNA expression data of 372 patients with their corresponding clinical characteristics. The authors identified a prognostic signature consisting of seven ferroptosis-related genes (*MAPK9*, *SLC1A4*, *PCK2*, *ACSL3*, *STMN1*, *CDO1*, and *CXCL2*) that was independently associated with poor DFS, and they employed this signature to construct a reliable risk model of ER, which was validated in an independent cohort of The Cancer Genome Atlas.

Table 5 summarizes findings from studies assessing the relationship between gene signatures and prognosis after LR for HCC.

Autophagy is a process that leads to the lysosome-mediated degradation of damaged or aging cellular components. Autophagy is pivotal in maintaining tissue homeostasis and is believed to play a role in tumorigenesis, cancer progression, and resistance to systemic therapy. [63] The prognosis after LR for HCC seems to be associated with autophagy-related genes. In particular, recent evidence showed how underexpression of autophagic genes leads to HCC recurrence and poor OS [72,73,74,75].

Notably, Lin et al. [73] assessed the expression of LC3, Beclin-1, and p62 in HCC and adjacent healthy tissue in 535 HCC patients who underwent radical hepatectomy. The authors performed a multivariate analysis and found that both early recurrence and late recurrence were associated with LC3 underexpression in the entire tumor microenvironment. In this study, HCC patients with low LC3 expression had a 5-year recurrence rate of 94.3%. The same research group also proved that a combination of Axl overexpression and LC3 underexpression in HCC patients who underwent LR was associated with the highest risk of recurrence (90% at 5 years) and mortality (83.2% at 5 years) [74]. 

Table 6 summarizes findings from studies assessing the relationship between autophagy-related genes and prognosis after LR for HCC.

## 4. Inflammation Markers

Several recent studies have highlighted a close correlation between inflammatory status, nutritional status, and the prognosis of HCC patients after curative-intent hepatectomy. Specifically, prognostic nutritional index (PNI), sarcopenia status, and several blood cell ratios have been adopted as independent predictors of OS [76,77] (Figure 3).

The role of the immune system in HCC development and progression is difficult to comprehend. Chronic inflammation is common in liver disease and plays a pivotal role in tumor cells’ ability to elude immune surveillance [78]. To further investigate the role of inflammation in HCC, Wu et al. [79] analyzed the NF-κB signal pathway in tumor specimens and correlated HCC progression with the inflammatory state of hepatitis. They showed that high levels of TNFα in the tumor microenvironment promote EMT by upregulating the transcriptional regulator Snail, promoting tumor invasion, and indirectly reducing DFS. In contrast, the presence of high levels of tumor-associated macrophages and memory T cells appeared to protect HCC patients from recurrence after hepatectomy, improving DFS and OS [80].

Several other studies showed that preoperative white blood cell counts and liver function test results were reproducible predictors of DFS and OS. Specifically, neutrophil-to-lymphocyte ratio, platelet-to-lymphocyte ratio, and aspartate aminotransferase-to-platelet count ratio index were found to be independent risk factors for poor outcomes in HCC patients after LR [81,82,83,84,85,86]. In a multicenter study [87], the aforementioned markers and additional preoperatively measured inflammation markers (aspartate aminotransferase-to-alanine aminotransferase ratio, aspartate aminotransferase-to-lymphocyte ratio index, and aspartate aminotransferase-to-neutrophil ratio index) were used to create an inflammation score system. A multivariate Cox analysis showed that a high inflammation score was an independent predictor of worse DFS and OS in HCC patients who underwent hepatectomy. This principle appeared to be significant for patients with early-stage HCC.

Notable new indexes include the following:

- The systemic immune-inflammation index, which utilizes lymphocyte, neutrophil, and platelet counts [88] to predict OS. A systemic immune-infiltration index ≥330 was significantly associated with ER after LR and higher CTC blood levels.

- The monocyte-to-lymphocyte ratio, which was shown to be a better predictor than systemic immune-infiltration index, neutrophil-to-lymphocyte ratio, or platelet-to-lymphocyte ratio for ER and DFS [89]. Furthermore, the monocyte-to-lymphocyte ratio combined with tumor size, tumor differentiation, and the Barcelona Clinic Liver Cancer stage appeared to be more reliable than the monocyte-to-lymphocyte ratio by itself.

Pinato et al. [90] first demonstrated how PNI was an independent predictor of OS in HCC patients, showing that nutritional status was strictly related to HCC progression. In particular, the 5-year DFS and OS rates in HCC patients with PNI < 45 were significantly lower than those in HCC patients with higher PNI, and low PNI was an independent outcome predictor for patients with early-stage HCC after curative-intent hepatectomy [91,92].

A meta-analysis by Man et al. [93], including 13 articles and 3738 HCC cases, proved that preoperatively low levels of PNI were statistically related to postoperative early recurrence and worse OS. Furthermore, a multicenter retrospective study encompassing both nutritional and inflammatory factors established that detection of sarcopenia on computed tomography and/or a high platelet-to-lymphocyte ratio is independently associated with poor OS in HCC patients undergoing LR [77].

Table 7 summarizes findings from studies assessing the relationship between inflammation markers and prognosis after LR for HCC.

## 5. Other Markers

In addition to the markers discussed above, a few other markers have been identified that are associated with prognosis after LR for HCC. These include PIVKA-II, immune checkpoint molecules, microRNAs, and proteins in urinary exosomes.

PIVKA-II is frequently used in combination with AFP to diagnose HCC, but the presence of PIVKA-II was recently proven to be an independent predictor of ER after LR. Wang et al. [8] observed 751 HCC patients undergoing LR and assessed serum AFP and PIVKA-II levels in the perioperative period and within 2 years after surgery. They demonstrated that PIVKA-II had a higher positivity rate than AFP in patients with ER. Moreover, unlike AFP, on multivariate Cox regression analysis, PIVKA-II was an independent predictor of ER after curative-intent hepatectomy.

The checkpoint molecule programmed cell death protein-1 (PD-1) has been proven to correlate with tumor aggressiveness. Shi et al. [94] conducted a study that included 56 hepatitis B virus-infected patients with HCC, 54 of whom underwent LR. These authors found high levels of T cells positive for both PD-1 and CD8 in peripheral blood samples of patients with advanced HCC, suggesting a positive correlation with HCC progression. Additionally, circulating PD-1 was found to be promising in predicting shorter DFS [94]. Nie et al. [95] utilized flow cytometry to detect programmed cell death-ligand 1 (PD-L1) levels in blood specimens of HCC patients undergoing radical hepatectomy and performed a multivariate Cox regression analysis, which showed that PD-L1 expression was one of the most significant independent predictors of DFS.

Researchers are searching for circulating microRNAs for diagnosis and prediction of prognosis in various liver diseases, including viral hepatitis, cirrhosis, and HCC [96]. Dysregulated microRNA expression and protein production are crucial to cancer initiation and malignant progression in some cancers. Chen et al. [97] reported that miR-182 and miR-331-3p might be useful in predicting the TNM stage and progression of HCC and are correlated with postoperative OS. For patients with hepatitis B virus-related HCC who undergo LR, Cho et al. [98] support circulating levels of miR-26a and miR-29a as predictors of poor DFS and OS. Recently, Wong et al. [10] combined quantitative PCR and artificial intelligence to integrate circulating microRNA with laboratory tumor characteristics and create a diagnostic and predictive model of DFS in patients undergoing LR. These authors identified a panel of eight microRNAs (which they termed the HCC seek-8 panel) that, in combination with serum biomarkers, was significantly associated with DFS.

Recently, several studies have focused on assessing the role of the exosomal proteome as a potential source of biomarkers for HCC [11,99]. Feng et al. [11] used supramolecular probe-based capture and in situ detection technology to demonstrate how exosomes are efficiently enriched from urine samples with high concentration and purity. The urinary exosome proteomic analysis identified 68 upregulated proteins in HCC patients. Additionally, three previously identified biomarkers, OLFM4, HDGF, and GDF15, were validated as diagnostic biomarkers for HCC, showcasing the ability to diagnose HCC and potentially recognize the ER of HCC in a noninvasive, reproducible, and reliable manner.

Table 8 summarizes findings from studies assessing the relationship between other markers and the prognosis after LR for HCC.

## 6. Conclusions

During the past decade, researchers have increased their effort to identify new biomarkers able to reliably predict survival and recurrence in HCC patients undergoing surgical resection. While some of the proposed markers have been widely investigated and demonstrated to be reproducible, others seem less reproducible in clinical practice.

Liquid biopsy has been proven to be a powerful tool to predict long-term outcomes of patients with resected HCC, with costs related to its technical implementation representing the main limitation. Inflammation markers, including PNI and several blood cell ratios, seem more affordable on a large scale. While numerous groups have tried to identify patterns of gene expression capable of predicting the survival of patients undergoing curative-intent hepatectomy for HCC, there is still too much heterogeneity in the findings for gene expression to be used in clinical practice.

It is important to note that the great majority of the available studies on this topic come from Eastern Asia, reflecting the incidence of HCC in the world. It would be important to obtain more evidence from Western countries in the future to confirm that findings from Eastern countries hold true for patients in Western countries.

## Figures and Tables

**Figure 1 cancers-16-02183-f001:**
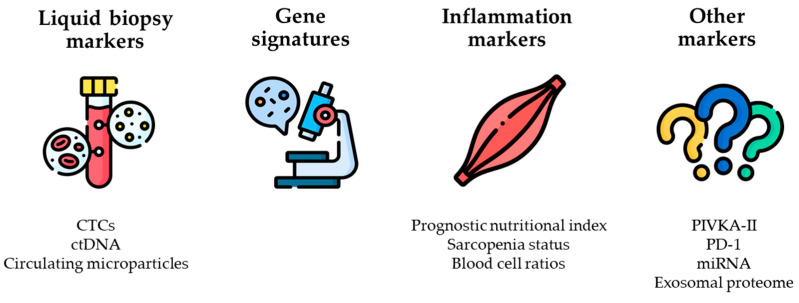
Categories of new prognostic factors in patients undergoing liver resection for hepatocellular carcinoma. Abbreviations: CTCs, circulating tumor cells; ctDNA, circulating tumor DNA; PIVKA-II, prothrombin induced by vitamin K absence-II; PD-1, programmed cell death protein-1; miRNA, microRNA. This figure has been designed using images from Flaticon.com.

**Figure 2 cancers-16-02183-f002:**
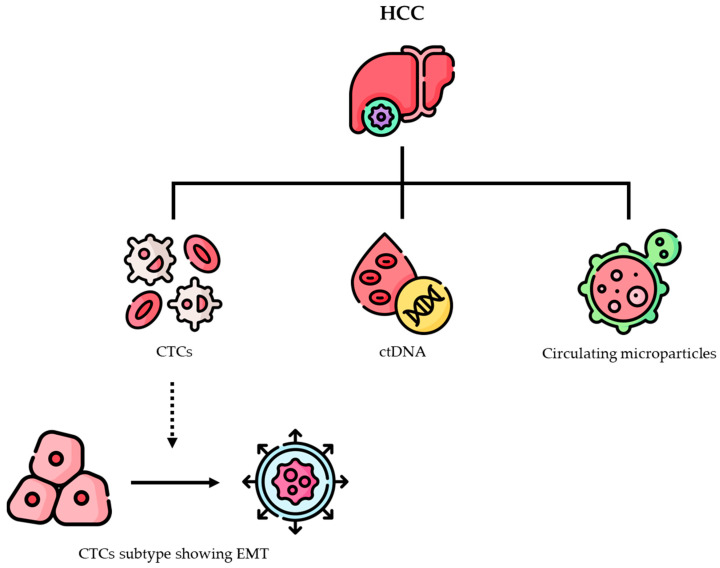
Liquid biopsy is the collection of tumor material from body fluids, such as circulating tumor cells, circulating tumor DNA, and circulating microparticles secreted by viable hepatocellular carcinoma cells. Abbreviations: HCC, hepatocellular carcinoma; CTCs, circulating tumor cells; ctDNA, circulating tumor DNA; EMT, epithelial-mesenchymal transition. This figure has been designed using images from Flaticon.com.

**Figure 3 cancers-16-02183-f003:**
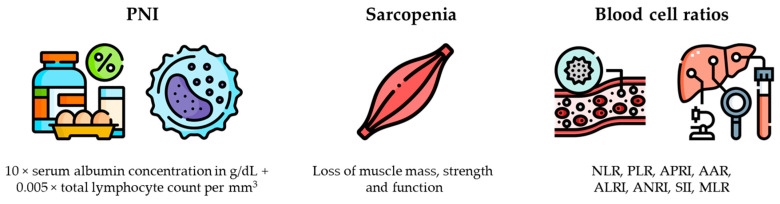
Prognostic nutritional index, sarcopenia status, and several blood cell ratios have been adopted as independent predictors of survival after resection of hepatocellular carcinoma. Abbreviations: PNI, prognostic nutritional index; NLR, neutrophil-to-lymphocyte ratio; PLR, platelet-to-lymphocyte ratio; APRI, aspartate aminotransferase-to-platelet count ratio index; AAR, aspartate aminotransferase-to-alanine aminotransferase ratio; ALRI, aspartate aminotransferase-to-lymphocyte ratio index; ANRI, aspartate aminotransferase-to-neutrophil ratio index; SII, systemic immune-inflammation index; MLR, monocyte-to-lymphocyte ratio. This figure has been designed using images from Flaticon.com.

**Table 1 cancers-16-02183-t001:** Studies assessing the relationship between CTCs and prognosis after LR for HCC.

Author	Year	Country	No. Patients	Method	Findings
Sun [18]	2013	China	123	CellSearch	Preoperative CTC count ≥ 2 CTCs/7.5 mL predicted HCC recurrence after LR, especially when AFP level was ≤400 ng/mL or tumor recurrence risk was low
Sun [19]	2020	China	197	CellSearch	Postoperative HCC CTC count ≥ 3 CTCs/7.5 mL predicted extrahepatic metastases and reduced survival after curative intent LR
Von Felden [20]	2017	Germany	61	CellSearch	Patients with EpCAM+ HCC CTCs demonstrated a higher risk of ER and lower DFS
Yu [21]	2018	China	139	CellSearch	Patients with preoperative CTC count < 2 CTCs/7.5 mL and postoperative CTC count ≥ 2 CTCs/7.5 mL had shorter DFS and OS
Amado [22]	2021	Spain	10	IsoFlux system	Unsatisfactory decrease in HCC CTC count after LR was associated with increased risk of recurrence and death
Fan [23]	2011	China	82	Multicolor flow cytometry	CTC percentage > 0.01% predicted intrahepatic and extrahepatic recurrence and was associated with worse OS after LR
Hamaoka [24]	2019	Japan	85	Multicolor flow cytometry	CTC count ≥ 5 CTCs/8 mL was associated with higher risk of microscopic portal vein invasion and lower OS. DFS rate decreased with increase in CTC number
Ha [25]	2019	South Korea	105	Tapered slit platform	CTC increase after LR predicted recurrence in patients with early-stage HCC after LR and was related to lower OS among patients with cirrhosis

Abbreviations: CTC, circulating tumor cell; LR, liver resection; HCC, hepatocellular carcinoma; AFP, alpha-fetoprotein; EpCAM+, epithelial cell adhesion molecule positive; ER, early recurrence; DFS, disease-free survival; OS, overall survival.

**Table 2 cancers-16-02183-t002:** Studies assessing the relationship between CTC subtypes and prognosis after LR for HCC.

Author	Year	Country	No. Patients	Method	Findings
Wang [29]	2018	China	62	Multiplex fluorescence in situ hybridization	The presence of mesenchymal CTCs was significantly correlated with DFS. Mesenchymal CTC positivity was an independent risk factor for ER
Ou [30]	2018	China	165	CanPatrol	Mesenchymal CTC phenotype is correlated with a higher AFP level, multiple tumors, advanced TNM and BCLC stages, the presence of emboli or microemboli, and ER
Yin [31]	2018	China	33	CanPatrol	A positive ratio of Twist+ CTCSs/CTCs was associated with the development of metastases, recurrence, and OS
Qi [32]	2018	China	112	CanPatrol	CTC count ≥ 16 CTCs/5 mL and mesenchymal-CTC percentage ≥ 2% prior to LR predicted ER and was associated with multiple intrahepatic recurrences, and lung metastases
Qi [33]	2020	China	136	CanPatrol	A high pre-resection CTC count the and presence of CTCs with mesenchymal and epithelial/mesenchymal phenotypes were significantly associated with extrahepatic recurrence and multiple intrahepatic recurrences
Zhang [34]	2021	China	105	CanPatrol	A high CTC count and a high percentage of mesenchymal CTCs were related to the expression of cytokeratin 19, which was associated with poor survival
Chen [35]	2019	China	143	CanPatrol	No correlation was noted between CTC count, mesenchymal phenotype, clinical stage, or ER
Court [36]	2018	USA	61	NanoVelcro CTC Assay	Vimentin-positive CTCs discriminated between patients with early-stage HCC eligible for liver transplant and patients with locally advanced/metastatic HCC ineligible for liver transplant and predicted OS for all patients

Abbreviations: CTC, circulating tumor cell; LR, liver resection; HCC, hepatocellular carcinoma; DFS, disease-free survival; ER, early recurrence; AFP, alpha-fetoprotein; BCLC, Barcelona Clinic Liver Cancer; Twist+, Twist-positive; OS, overall survival.

**Table 3 cancers-16-02183-t003:** Studies assessing the relationship between ctDNA and ctDNA mutations and prognosis after LR for HCC.

Author	Year	Country	No. Patients	Method	Findings
Tokuhisa [45]	2007	Japan	96	RT-PCR assay with *GSTP1*	ctDNA level was an independent predictor of OS and extrahepatic metastases
An [46]	2019	China	26	Qiagen	Postoperative ctDNA mutations were an independent predictor of ER
Cai [47]	2019	China	34	Qiagen	Serial ctDNA sampling provided optimum performance in HCC surveillance, diagnosing microscopic residual tumors and discovering recurrence before imaging
Liao [48]	2016	China	41	MiSeq	HCC ctDNA mutations are correlated with vascular invasion and shorter DFS
Shen [49]	2020	China	275	Droplet digital PCR	Among patients with resected HCC, patients with the *TP53* R249S mutation in ctDNA had worse OS and DFS than patients without this mutation
Liu [50]	2017	China	75	TIANGEN and Axygen	*LINE1* hypomethylation and *RASSF1A* promoter hypermethylation predicted poor OS and ER
Chan [51]	2013	China	32	AMPure XP magnetic beads and Qiagen	Residual HCC after LR was related to the extent of hypomethylation in cell-free DNA
Xu [52]	2017	China/USA	434	Illumina sequencing	A 10-marker panel able to predict prognosis was developed, and a prognostic score was created (cd-score); cd-scores were higher before surgery than after surgery and higher in patients with recurrence than in those without recurrence

Abbreviations: ctDNA, circulating tumor DNA; LR, liver resection; HCC, hepatocellular carcinoma; RT-PCR, reverse-transcription polymerase chain reaction; OS, overall survival; Qiagen, QIAamp circulating nucleic acid kit; ER, early recurrence; MiSeq, Illumina MiSeq benchtop sequencing system; DFS, disease-free survival; TIANGEN, TIANGEN TIANamp kit; Axygen, Axygen body fluid viral DNA/RNA miniprep kit; Illumina, Illumina clonal amplification and sequencing by synthesis chemistry.

**Table 4 cancers-16-02183-t004:** Studies assessing the relationship between CMs and prognosis after LR for HCC.

Author	Year	Country	No. Patients	Method	Findings
Abbate [55]	2017	Italy	15	Flow cytometry	HepPar1+ CM concentration before LR was higher in HCC patients with ER than in those with no recurrence
Liu [56]	2017	China	218	ExoQuick exosome precipitation solution and qRT-PCR	Low exosomal miR-125b levels were associated with a higher risk of recurrence and worse OS
Shi [57]	2018	China	126	SYBR Green mastermix kit and Qiagen	Downregulation of miR-638 predicted a poor prognosis
Tian [58]	2019	China	124	miRNA microarray	miR-21 and miR-10b in HCC promote cancer cell proliferation and metastasis. In patients with early-stage HCC undergoing LR, exosomal miR-21 and miR-10b expression were independent risk factors for short-term DFS
Luo [59]	2020	China	124	qRT-PCR	circAKT3 is higher in subjects with ER and is related to poor OS

Abbreviations: CMs, circulating microparticle; LR, liver resection; HCC, hepatocellular carcinoma; HepPar1+, hepatocyte paraffin 1–positive; ER, early recurrence; qRT-PCR, quantitative-reverse-transcription polymerase chain reaction; Qiagen, QIAamp circulating nucleic acid kit; miRNA, micro-RNA; DFS, disease-free survival; circAKT3, circular RNA AKT3; OS, overall survival.

**Table 5 cancers-16-02183-t005:** Studies assessing the relationship between gene signatures and prognosis after LR for HCC.

Author	Year	Country	No. Patients	Method	Findings
Ashida [65]	2017	Japan	92	Qiagen miRNeasy Mini Kit, RT-PCR, and RNA microarray	Down-regulation of the *CYP3A4* gene was an independent predictor of ER and OS
Wang [66]	2019	China	64	RT-PCR and RNA microarray	*NUF2* expression in combination with liver cirrhosis reliably predicted ER and poor OS
He [67]	2021	China	297	NGS	Gene rearrangement was associated with *TP53* mutations and worse DFS
Hwang [68]	2021	South Korea	206	Gene Expression Omnibus	*S100P* was an independent predictor of postresection OS
Song [69]	2021	China	183	NGS	*TSC2* mutations were independently associated with recurrence within 1 year and poorer DFS
Wang [70]	2022	China	372	*Homo sapiens* and expression profiling by array	A seven-gene signature based on *MAPK9, SLC1A4, PCK2, ACSL3, STMN1, CDO1,* and *CXCL2* was independently associated with poor DFS
Son [61]	2022	South Korea	85	RNA microarray	The combination of *HMGA1* and *MPZL1* upregulation is an excellent predictor of HCC recurrence. *HMGA1* and *RACGAP1* might be used as independent prognostic factors in patients with HCC
Xin [71]	2022	China	41	Qiagen	*KEAP1*, *TP53*, *H3C4* (previously *HIST1H3D*), *NFKBIA*, *PIK3CB*, and *WRN* mutations have a higher incidence in patients with ER than in patients without ER

Abbreviations: LR, liver resection; HCC, hepatocellular carcinoma; Qiagen, QIAamp circulating nucleic acid kit; RT-PCR, reverse-transcription polymerase chain reaction; ER, early recurrence; OS, overall survival; NGS, next-generation sequencing; DFS, disease-free survival.

**Table 6 cancers-16-02183-t006:** Studies of the relationship between autophagy-related genes and prognosis after LR for HCC.

Author	Year	Country	No. Patients	Method	Findings
Ding [72]	2008	China	300	RT-PCR, Western blotting	Underexpression of the autophagic gene *BECN1*, associated with overexpression of the antiapoptotic gene BCL2L1, significantly correlates with DFS and OS
Lin [73]	2018	Taiwan	535	IHC	High LC3 expression in tumor and liver microenvironments is significantly associated with a lower rate of HCC recurrence
Hsu [74]	2019	Taiwan	535	IHC	High Axl expression in HCC is associated with aggressive tumor behavior and worse clinical outcomes. The combination of high Axl expression and low LC3 expression significantly predicts a poorer prognosis after LR
Wang [75]	2023	China	29	Not reported	A gene signature composed of the autophagy-related genes *CLN3*, *HGF*, *TRIM22*, *SNRPD1*, and *SNRPE* is an independent risk factor for poor DFS

Abbreviations: LR, liver resection; HCC, hepatocellular carcinoma; RT-PCR, real-time polymerase chain reaction; DFS, disease-free survival; OS, overall survival; IHC, immunohistochemistry.

**Table 7 cancers-16-02183-t007:** Studies assessing the relationship between inflammation markers and prognosis after LR for HCC.

Author	Year	Country	No. Patients	Method	Findings
Mao [76]	2022	China	360	Blood test	PNI and NLR are independent prognostic factors for OS. APRI is an independent prognostic factor for DFS, 1-year DFS, and 2-year DFS. SIRI is an independent prognostic factor for 1-year DFS in HCC patients after LR
Kim [77]	2022	South Korea	159	Blood test	Sarcopenia and a high PLR are significant predictors of OS
Wu [79]	2016	Taiwan	232	Cytokine array	High levels of TNFα in the tumor microenvironment may promote EMT by upregulating the transcriptional regulator Snail, promoting tumor invasion, and reducing DFS
Li [80]	2009	China	302	IHC	High TAM infiltration is associated with improved DFS and OS. TAM infiltration is an independent prognostic factor for DFS and OS. The synergic effect of TAM and memory T cell infiltration influences the DFS and OS of HCC patients after LR
Mano [81]	2013	Japan	958	Blood test	NLR is an independent predictor of survival after LR in patients with HCC
Shen [82]	2014	China	332	Blood test	DFS and OS are significantly better in patients with a low platelet count, PLR, and APRI than in patients with elevated values
Ji [83]	2016	China	321	Blood test	Preoperative NLR and APRI are independent predictors of DFS and OS. A higher NLR or APRI predicts poorer outcomes in HCC patients
Liu [84]	2016	China	223	Blood test	NLR > 2.75 and APRI > 0.23 are independent adverse prognostic factors for ER
Hu [85]	2016	South Korea	213	Blood test	NLR ≥ 1.945 was a prognostic factor for ER
Wang [86]	2019	China	239	Blood test	NLR ≥ 2.92 and PLR ≥ 128.1 are prognostic factors for poor outcomes after LR
Chen [87]	2020	China	455	Blood test	A high inflammation score, including AAR, ALRI, PLR, NLR, and ANRI, is an independent risk factor for worse OS and RFS
Hu [88]	2014	China	256	Blood test	SII is an independent predictor for OS and DFS
Wu [89]	2021	China	161	Blood test	The combination of MLR and clinical risk factors is helpful to identify patients with HCC at high risk for ER
Fan [91]	2021	China	187	Blood test	PNI < 45 is associated with poor DFS. Preoperative PNI is an independent prognostic factor for OS and DFS
Chan [92]	2015	China	324	Blood test	PNI < 45 correlates with adverse OS and DFS. It is an independent predictor of disease-specific death and early and late tumor relapses
Man [93]	2018	China	3738	Blood test	Preoperative PNI is a prognostic marker in resected HCC

Abbreviations: LR, liver resection; HCC, hepatocellular carcinoma; PNI, prognostic nutritional index; NLR, neutrophil-to-lymphocyte ratio; OS, overall survival; APRI, aspartate aminotransferase-to-platelet count ratio index; DFS, disease-free survival; SIRI, systemic inflammation response index; PLR, platelet-to-lymphocyte ratio; TNFα, tumor necrosis factor alpha; EMT, epithelial-mesenchymal transition; IHC, immunohistochemistry; TAM, tumor-associated macrophages; ER, early recurrence; AAR, aspartate aminotransferase-to-alanine aminotransferase ratio; ALRI, aspartate aminotransferase-to-lymphocyte ratio index; ANRI, aspartate aminotransferase-to-neutrophil ratio index; RFS, recurrence-free survival; SII, systemic immune-inflammation index; MLR, monocyte-to-lymphocyte ratio.

**Table 8 cancers-16-02183-t008:** Studies assessing the relationship between other markers and prognosis after LR for HCC.

Author	Year	Country	No. Patients	Method	Findings
Wang [8]	2022	China	751	Blood test	AFP and PIVKA-II predicted ER after HCC resection. Preoperative PIVKA-II positivity is independently associated with ER after LR
Shi [93]	2011	China	54	Flow cytometry	An increase in circulating and intratumor PD-1+ CD8+ T cells could predict poorer DFS and recurrence
Nie [94]	2021	China	72	Flow cytometry	Expression of CD8/CD3 and PD-L1 in tumor-infiltrating lymphocytes of HCC patients represents a prognostic factor after radical resection
Chen [96]	2015	China	103	qRT-PCR	Serum miR-182 and miR-331-3p were associated with the postoperative survival of HCC patients, and both markers were shown to be independent prognostic factors
Cho [97]	2017	South Korea	63	qRT-PCR	Pretreatment levels of circulating miR-26a and miR-29a are independent prognostic markers for poor DFS in patients with hepatitis B virus-related HCC
Wong [10]	2023	China	103	qPCR and machine learning	An 8-microRNA panel (HCCseek-8 panel) was identified and proved to be correlated with DFS. The same panel in combination with AFP, AST, and ALT was associated with even better prediction power
Feng [99]	2023	China	18	ADSP array technology	Sixty-eight proteins were up-regulated in the urinary exosomes of HCC patients. OLFM4, HDGF, and GDF15 were validated as HCC diagnostic biomarkers on the basis of proteomic expression

Abbreviations: LR, liver resection; HCC, hepatocellular carcinoma; AFP, alpha-fetoprotein; PIVKA-II, prothrombin induced by vitamin K absence-II; ER, early recurrence; PD-1+, programmed cell death protein-1 positive; CD8+, CD8 positive; DFS, disease-free survival; PD-L1, programmed death-ligand 1; qRT-PCR, quantitative-reverse-transcription polymerase chain reaction; AST, aspartate aminotransferase; ALT, alanine aminotransferase; ADSP, array-based amphiphilic supramolecular probe.

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
