# Peer review of "Emerging Prognostic Markers in Patients Undergoing Liver Resection for Hepatocellular Carcinoma: A Narrative Review"

_cancers, 2024, doi:10.3390/cancers16122183_

Round 1

Reviewer 1 Report (Previous Reviewer 3)

Comments and Suggestions for Authors

In the present work, Panettieri et al. try to review the new prognostic factors in patients undergoing liver resection for hepatocellular carcinoma. There are some questions that should be explained.

Major concerns

1. The manuscript should be supported by a professional English language proofreading service.

2. The article should be rewritten based on the style of Good Review papers.

Specific comments

1. Abstract section

Lines 17-19, ‘Pathologic factors (i.e.: microvascular and capsular invasion) and alpha-fetoprotein (AFP) increase have been traditionally recognized as predictors of recurrence and survival.’ These sentences should be rewritten.

Delete ‘(AFP)’ and ‘(PNI)’.

Although a conclusion in Abstract section has be added, but it should revise in the style of ‘In this Review, we discuss…..’.

Keywords, ‘liver resection’ and ‘hepatectomy’ are the same meaning.

2. Introduction section

The style of reference citation should be revised based on this Journal, including the format of references that is not suitable for this Journal.

Figure 1 should present in the text.

Lines 57-58, ‘Unfortunately, in this increasing landscape, not all the aforementioned markers are easily available in clinical practice, and their actual reproducibility is unclear.’ These sentences should be rewritten.

Lines 58-60, ‘The aim of this review is to summarize the use of emerging non-invasive biomarkers in predicting survival outcomes after LR for HCC.’ In order to what?

5. Liquid biopsy section

Lines 113, 168, ‘HCC CTCs’.

Lines 131-134, there is only one reference for this short paragraph.

Lines 153-155, there is only one reference for this short paragraph.

Lines 156-161, there is only one reference for this short paragraph.

Lines 198-202, there is only one reference for this short paragraph.

There are so many short paragraphs with only one reference.

6. As a review article, there are many exact data, including p value. The conclusion of the cited article is reviewed only, and narrating exact data is not necessary.

7. There are four short paragraphs in Conclusion section, which should be refined.

Comments on the Quality of English Language

 Extensive editing of English language required.

Author Response

We thank reviewer #1 for their insightful comments and suggetsions. We have edited the manuscript accordingly.

  • The article has been completely revised and rewritten by the senior scientific editor at The University of Texas, MD Anderson Cancer Center (Ms. Stephanie Deming), as acknowledged in the manuscript. PRISMA guidelines should be followed when writing systematic reviews and meta-analyses. They are generally not a requirement for narrative reviews.  
  • The abstract, keywords, and references style have been reviewed. 
  • References style has been reformatted. 
  • Figure 1 was added to the text. 
  • Introduction has been improved as suggested. 
  • Exact datas have been reduced, eliminating p values. 
  • The conslusion section has been redifined. 

As previously stated, the manuscript has been improved and completely rewritten thanks to a professional copy-editing service. For further questions Ms. Deming can be contacted at sdeming@mdanderson.org.

We remain available if reviewers should require additional changes.

Reviewer 2 Report (New Reviewer)

Comments and Suggestions for Authors

This narrative review manuscript was well organized. The contents of the manuscript can provide recent important information to the readers. I recommend publishing this manuscript.

Minor comment:

The sentences from line 297-299 [ GR-HCCs were 45.1%, 31.9%, 31.9% at 1-, 2-, and 3- years after 297 LR, respectively, and when significantly worse when compared to those without GR (72.5%, 57.9%, and 49.0%, respectively; p = 0.001). Furthermore, GR appeared to be an in- 299 dependent risk factor for lower DFS.] were hard to be understanded.

The meaning in original reference were: The 1-, 2-, and 3-year cumulative DFS rates in GR-HCCs were 45.1%, 31.9%, 31.9%, respectively, which were significantly lower than those of GR-negative HCCs (NGR-HCCs) (72.5%, 57.9%, and 49.0%, respectively; P = 0.001). GR was identified as an independent risk factor for inferior DFS in HCCs (HR = 1.980, 95% CI = 1.246-3.147; P = 0.004).

Author Response

We thank reviewer #2 for their comments and we edited the manuscript based on their suggestions. 

Round 2

Reviewer 1 Report (Previous Reviewer 3)

Comments and Suggestions for Authors

Thanks for author’s responses. The language corrections had been performed throughout the paper, and the format or style of all references had been corrected. However, as a high impact factor Journal for a review paper, some Figures should be added, for example, the origins of prognostic markers, and others.

Comments on the Quality of English Language

Minor editing of English language required.

Author Response

We thank reviewer #1 for their suggestion, and we added some additional figures, as requested.

This manuscript is a resubmission of an earlier submission. The following is a list of the peer review reports and author responses from that submission.

Round 1

Reviewer 1 Report

Comments and Suggestions for Authors

The manuscript described the current status for liver cancer biomarker discovery and the potential clinical translational study. Although AFP can only account for 70% accuracy, it’s still considered as the standard for diagnosis. Therefore, the novel markers are of great importance but also need long term validation. In this review, the authors also talked about the exosome-derived biomarkers, which is the next generation source since the bilayer protect the intra-molecules to be highly stable. However, the manuscript focused more on the miRNA instead of proteins. Actually the proteomics study has push more proteins biomarkers to be discovered which is highly related with tumor(J Proteome Res. 2023;22(7):2516-2524). Also the glycosylation forms of proteins also serve as important diagnosis markers as described. I would suggested the authors include the proteomic study of exosome as part of this review. I would like to recommend these two references: ACS Omega. 2021 Jan 4;6(1):827-835 and J Proteome Res. 2023 Jul 7;22(7):2516-2524.

Author Response

We thank Reviewer #1 for their valuable advice and we have modified the paper based on their recommendations. 

The manuscript has been edited as follows:

"Recently, several studies focused on assessing the role of the urinary exosomal proteome as a potential source of biomarkers for HCC. [101,102] Feng et al. [102] used a supramolecular probe-based capture and in situ detection technology to demonstrate how exosomes are efficiently enriched from urine samples with high concentration and purity. The urinary exosome proteomic analysis identified 68 up-regulated proteins in HCC patients. Additionally, OLFM4, HDGF, and GDF15 – three proteins whose biomarker value has already been reported - were validated, showcasing the potential of this approach to diagnose HCC and potentially recognize its ER in a non-invasive, reproducible, and reliable manner".

Table 8 has been updated accordingly. 

Reviewer 2 Report

Comments and Suggestions for Authors

This review entitled "New prognostic factors in patients undergoing liver resection for hepatocellular carcinoma: what is applicable in clinical practice? A narrative review". is outstanding, timely, and well written. It will be germane to health providers, scientists and above all to oncologists. The authors stress that there is an urgent need to find novel non-invasive predictors that  drive medical and surgical treatments. three categories are discussed: liquid biopsy markers,pathology markers and inflammatory markers. They conclude  that liquid biopsy has become an established tool for the prediction of long-term outcomes of resected HCC patients. Genetic mutations  are evolving,

A major strength of the review is much of the data comes from Asia where HCC is endemic. They are asking for HCC data from many other countries to assess their experiences and whether it is similar or different.

In your interactions with other centers, countries and investigators how many are getting government financial support? 

Author Response

We thank reviewer #2 for their appreciation and support. 

Based on our interactions and connections, among Western countries, government financial support is more easily available in North America than in Europe. In particular, American investigators can conduct NIH funded research. 

Reviewer 3 Report

Comments and Suggestions for Authors

In the present work, Panettieri et al. try to review the new prognostic factors in patients undergoing liver resection for hepatocellular carcinoma. There are some questions that should be explained, and some new and related references should be added.

1. A conclusion in Abstract section should be added.

2. Lines 59-64, these two paragraphs may be amalgamated into one paragraph.

3. Line 111, ‘(p < 0.05). (Table 1).’ should be revised.

4. Tables 1-8, some ‘Methods’ should be revised, and all ‘Aims’ should be revised, and all ‘Findings’ should be refined.

5. In this review article, there are many exact data, including p value, which may be not suitable.

6. Some summary Figures may be needed in this review.

7. Lines 365-395, there are so many paragraphs, and some paragraphs may be amalgamated into one paragraph.

8. References, Format of references is not suitable for this Journal.

9. Some new references are related to this manuscript. Therefore, the novelty of this manuscript may be not enough.

Nevola R, Ruocco R, Criscuolo L, Villani A, Alfano M, Beccia D, Imbriani S, Claar E, Cozzolino D, Sasso FC, Marrone A, Adinolfi LE, Rinaldi L. Predictors of early and late hepatocellular carcinoma recurrence. World J Gastroenterol. 2023;29(8):1243-1260.

KaraoÄŸullarından Ü, Üsküdar O, OdabaÅŸ E, Ak N, Kuran S. Hepatocellular Carcinoma in Cirrhotic Versus Noncirrhotic Livers: Clinicomorphologic Findings and Prognostic Factors. Turk J Gastroenterol. 2023;34(3):262-269.

Huang J, Li L, Liu FC, Tan BB, Yang Y, Jiang BG, Pan ZY. Prognostic Analysis of Single Large Hepatocellular Carcinoma Following Radical Resection: A Single-Center Study. J Hepatocell Carcinoma. 2023;10:573-586.

Xu JX, Qin SL, Wei HW, Chen YY, Peng YC, Qi LN. Prognostic factors and an innovative nomogram model for patients with hepatocellular carcinoma treated with postoperative adjuvant transarterial chemoembolization. Ann Med. 2023;55(1):2199219.

Meng XQ, Miao H, Xia Y, Shen H, Qian Y, YanChen, Shen F, Guo J. A nomogram for predicting post-hepatectomy liver failure in patients with hepatocellular carcinoma based on spleen-volume-to-platelet ratio. Asian J Surg. 2023;46(1):399-404.

Comments on the Quality of English Language

Moderate editing of English language required.

Author Response

We thank reviewer #3 for their insightful comments.

Here is our point-by-point response. Changes have been marked in yellow in the manuscript. The English has been reviewed and improved. 

1) We added a conclusion to the abstract. 

2) The two paragraphs indicated by the reviewer have been merged into a single paragraph. 

3) Punctuation has been corrected as suggested.

4) We have refined the tables and made them easier to read. If needed, we are willing to further improve them, pending more accurate suggestions. 

5) We have followed the reviewer's suggestion and removed most of the exact data to make the manuscript a smoother read. 

6) We agree that a figure could help summarize the main categories of new prognostic factors after liver resection for HCC and we created one. 

7) The paragraphs have merged into one, as suggested. 

8) References have been reformatted. 

9) We thank the reviewer for their suggestion, nevertheless we are not sure the recommended papers are pertinent to our aim, which is to analyze the diffusion and actual utilization of novel prognostic markers, with a specific focus on results after liver resection. The review bu Nevola et al. describes all the possible HCC treatments and do not mention the noverl markers we illustrated. The papers from KaraoÄŸullarından and Huang are retrospective analyses of pathological findings of resected HCC specimens that do not add anything to the current knowlege. Xu et al. ficused on prognostic factors after TACE and Meng et al. analyzed prognotic factor for liver failure after liver resection. Both papers do not focus on long term survival after solely liver resection.